# The role of CDH2 and MCP-1 mRNAs of blood extracellular vesicles in predicting early-stage diabetic nephropathy

Hojat Dehghanbanadaki[1,2], Katayoon Forouzanfar[3], Ardeshir Kakaei[2], Samaneh Zeidi[1,4], Negar Salehi[1,4], Babak Arjmand[3,5], Farideh Razi[1,6]*, Ehsan Hashemi[1,4]*

1 Diabetes Research Center, Endocrinology and Metabolism Clinical Sciences Institute, Tehran University of Medical Sciences, Tehran, Iran, 2 Metabolomics and Genomics Research Center, Endocrinology and Metabolism Molecular–Cellular Sciences Institute, Tehran University of Medical Sciences, Tehran, Iran, 3 Metabolic Disorders Research Center, Endocrinology and Metabolism Molecular-Cellular Sciences Institute, Tehran University of Medical Sciences, Tehran, Iran, 4 Department of Animal Biotechnology, National Institute of Genetic Engineering and Biotechnology, Tehran, Iran, 5 Cell Therapy and Regenerative Medicine Research Center, Endocrinology and Metabolism Molecular–Cellular Sciences Institute, Tehran University of Medical Sciences, Tehran, Iran, 6 Endocrinology and Metabolism Research Center, Endocrinology and Metabolism Clinical Sciences Institute, Tehran University of Medical Sciences, Tehran, Iran

* F-razi@tums.ac.ir (FR); E_hashemi@nigeb.ac.ir (EH)

**Data Availability Statement:** All relevant data are within the paper and its Supporting information files.

## Abstract

### Background

Extracellular vesicles (EVs), including exosomes and microvesicles, are involved in intercellular communication by transferring biomolecules such as mRNA, which has been shown to be as essential biomarkers for many physiological and pathological conditions such as diabetic nephropathy (DN). This study aimed to investigate the expression of CDH1, CDH2, MCP-1, and PAI-1 mRNAs in blood EVs of DN patients and to determine their accuracy in predicting early-stage DN.

### Methods

We recruited 196 participants, including 35 overt DN patients, 53 incipient DN patients, 62 diabetic patients (DM), and 46 healthy individuals. Quantification of the mRNA profile of blood EVs was performed using the qRT-PCR method. The diagnostic performance of mRNA was evaluated using receiver operating characteristic analysis.

### Results

The mRNA expression of CDH2 and MCP-1 was downregulated in overt DN group (0.22-fold change and 0.15-fold change, respectively) and incipient DN group (0.60-fold change and 0.43-fold change, respectively) compared to DM group (1.72-fold change and 2.77-fold change, respectively), while PAI-1 mRNA expression decreased in incipient DN group (0.70-fold change) and DM group (0.58-fold change) compared to control. However, the expression level of CDH1 mRNA was not significantly different among the four groups ($p = 0.408$). Moreover, CDH2 and MCP-1 mRNAs inversely correlated with creatinine (r = -0.370

**Funding:** The authors received no specific funding for this work.

**Competing interests:** The authors have declared that no competing interests exist.

and r = -0.361, $p$<0.001) and Alb/Cr ratio (r = -0.355 and r = -0.297, $p$<0.001). 1/CDH2 mRNA also predicted overt DN with an accuracy of 0.75 (95%CI: 0.65–0.85) and incipient DN with an accuracy of 0.61 (95%CI: 0.50–0.71) while 1/MCP-1 mRNA had an accuracy of 0.66 (95%CI: 0.55–0.77) for overt DN prediction and an accuracy of 0.61 (95%CI: 0.51–0.71) for incipient DN prediction.

## Conclusion

CDH2 and MCP-1 mRNAs expression in blood EVs was decreased with the development of DN, suggesting the renoprotective effect of these mRNAs in diabetic individuals. Moreover, their quantifications could serve as diagnostic biomarkers for early-stage DN.

## Introduction

Diabetic nephropathy (DN) is one of the most common microvascular complications of diabetes mellitus (DM), occuring in almost one-third of diabetic patients [1] and is also a leading cause of renal failure worldwide, affecting 40% of patients with end-stage renal disease (ESRD) [2, 3]. Because diabetic patients suffering from DN are at higher risk of cardiovascular morbidity and mortality, the diagnosis and treatment of DN in the early stage has become a global public health issue [4, 5]. Currently, the clinical diagnosis of DN is based on the presence of albuminuria which lacks sufficient sensitivity and specificity for predicting early-stage DN and its progression. In addition, a number of studies suggest that renal dysfunction can occur even in the absence of microalbuminuria [6–9]. The gold standard for DN diagnosis is based on the histopathological findings of a renal biopsy, an invasive procedure with serious complications such as infection and bleeding. In addition, DN patients cannot be followed up with serial renal biopsy to monitor disease progression [10]. Therefore, accurate noninvasive surrogates that can be used to predict early-stage DN and monitor disease progression are currently needed. In this case, new sensitive laboratory biomarkers, especially in combination with conventional biomarkers, can accurately detect DN patients in its early-stage, stratify them into different stages for individualized management, and monitor their response to therapy [11]. To this end, extracellular vesicles (EVs) have emerged as promising candidates, and their alterations in abundance and composition are associated with various conditions, including DM and DN [12–15]. EVs are lipidic, spherical, nanosized vesicles that are actively produced by almost all cell types and released into various body fluids such as serum, plasma, amniotic fluid, saliva, and urine [16]. Exosomes and microvesicles are two major subtypes of EVs that differ in size, biogenesis, and expressed biomarkers from the cell of origin. Exosomes are 30–100 nm in diameter and originate from the endosomal pathway and fusion of multivesicular bodies with plasma membranes, whereas microvesicles, with a diameter of 100–1,000 nm, originate from budding and evagination of plasma membranes [17]. EVs were initially proposed to be debris of cells [18]. Nowadays, we know that EVs transport various biomolecular components such as metabolites, lipids, proteins, RNA, and DNA from the cells of origin to the target cells, contribute to intercellular communications and play an important role in homeostasis in physiological condition and disease development under pathological conditions [19]. Previous studies have shown that EVs may be good candidates for early detection and monitoring of DN [13, 14, 20–24]. In this case, exosomal Wilm's tumor-1 (WT1) mRNA from urine sediment of DN patients is inversely associated with estimated glomerular filtration rate (eGFR), indicating glomeruli damage and early renal failure. Moreover, this biomarker was found to

predict DN with an accuracy of 0.705 [24]. In addition, a trial with 9 participants showed that mRNA expression of cadherin 1 (CDH1), cadherin 2 (CDH2), monocyte chemoattractant protein-1 (MCP-1), plasminogen activator inhibitor-1 (PAI-1), and angiotensin I-converting enzyme (ACE) genes in the urine pellet was significantly increased in DN patients compared with healthy controls, and of these biomarkers, CDH2 mRNA was increased 15-fold [25]. CDH1, CDH2, MCP-1, and PAI-1 are protein-coding genes that are transcribed into their corresponding mRNAs for protein synthesis. In this case, CDH1 and CDH2 mRNAs produce calcium-dependent adhesion proteins (E-cadherin and N-cadherin), which play a fundamental role in intercellular adhesion, and abnormal expression of these genes causes various diseases such as DN, diabetic retinopathy, and epithelial-derived solid tumors [25–27]. The MCP-1 gene produces chemokines that play a pivotal role in regulating monocyte/macrophage trafficking in the inflammatory process. Abnormal expression of this gene leads to DN, diabetic retinopathy, insulin resistance, atherosclerosis, and other inflammatory conditions [28–30]. The PAI-1 gene, also known as serpin family E member 1 (SERPINE1), produces plasminogen activator inhibitor 1 that plays a major role in hemostasis by inhibiting plasminogen activators mediated fibrinolysis and in cell motility and adhesion by inhibiting cell migration-mediated by vitronectin and integrin. Abnormal PAI-1 gene expression leads to various diseases such as DN and cardiovascular disease due to DM [31–34]. Previous studies have shown that CDH1, CDH2, MCP-1, and PAI-1 mRNAs can act as potential predictors of renal fibrosis due to chronic kidney disease [35], obstructive nephropathy [36], and DN [37–39]. In this study, we aimed to investigate the expression levels of CDH1, CDH2, MCP-1, and PAI-1 mRNAs in blood EVs of overt DN patients, incipient DN patients, DM patients, and healthy controls and determine their diagnostic potential as novel biomarkers for predicting early-stage DN.

## Materials and methods

### Subjects and ethical considerations

A total of 196 participants, including 35 overt DN patients, 53 incipient DN patients, 62 DM patients, and 46 age- and sex-matched healthy controls, were recruited at an outpatient diabetes clinic between November 2018 and December 2019. All diabetic patients had type 2 diabetes mellitus and were divided into 3 subgroups according to the amount of albuminuria in three spot urine samples within 6 months. These included 62 diabetic patients with normoalbuminuria defined as urinary albumin excretion (UAE) < 30mg/gr creatinine in at least two samples (presented as DM group), 53 diabetic patients with microalbuminuria defined as UAE ≥ 30 and < 300 mg/gr creatinine in at least two samples (presented as incipient DN group), and 35 diabetic patients with macroalbuminuria defined as UAE ≥ 300 mg/ creatinine in at least two samples (presented as overt DN group). Participants who were younger than 30 years, older than 75 years, pregnant, or with comorbidities other than DM and DN such as cardiovascular disease, severe liver failure, urinary tract infection, and chronic inflammation were excluded. Besides, we included only participants with an eGFR ≥ 60 mL/min/1.73 m$^2$ because our research question concerned early-stage DN.

This study complied with all ethical guidelines for human experimentation in Helsinki Declaration and was approved by the Ethics Committee of Endocrine & Metabolism Research Institute and Tehran University of Medical Sciences (approval ID: IR.TUMS.EMRI.REC.1399.015). Written informed consent was obtained from all participants before participation.

### Sample collection and clinical laboratory tests

After at least 10 hours of fasting, 15 ml of venous blood and random urine were sampled from each participant. 5 ml of the blood sample was analyzed for routine clinical laboratory tests,

including urea, creatinine, fasting blood sugar (FBS), and HbA1c, while the remaining 10 ml was used for EVs extraction. The TOSOH G8 system was applied for HbA1c assessment, and a Roche commercial kit (Roche, Germany) was applied for the other laboratory tests. The Cockcroft–Gault equation was used to calculate eGFR [40] as follows:

$$\text{eGFR} = \frac{(140 - age) \times weight}{Serum\ creatinine\ (\frac{mg}{dl}) \times 72} \times 0.85\ \text{if woman.}$$

## Blood EVs extraction and assessment

The preparation and extraction process of blood EVs was described in a previous study [41]. Briefly, the plasma from 10 ml of blood was isolated (centrifugation, 290×g, 20 min, 4 ˚C) and centrifuged (12000×g, 20 min, 4 ˚C). Then, the supernatants were isolated and centrifuged (17000×g, 20 min, 4 ˚C). The obtained supernatants were picked up, and the pellets were suspended, centrifuged (10000×g,90 min, 4 ˚C), and resuspended in a phosphate-buffered sterile saline solution. The prepared pellets were stored at − 80 ˚C until RNA isolation.

The morphology and size of the isolated EVs were assessed by scanning electron microscopy (SEM) and dynamic light scattering (DLS). For SEM assessment, we fixed the isolated EVs in PBS for 1 hour with 2.5% glutaraldehyde and then coated them with a gold layer to improve electrical conductivity and finally examined the samples with VEGA TESCAN. For DLS assessment, we carefully added EVs suspensions to a cuvette and examined the size and density of EVs (Malvern zetasizer, Worcestershire, UK).

## mRNA isolation, cDNA preparation, and qRT-PCR analysis

According to the manufacturer's instructions, we isolated total RNA from EVs using TRIzolTM reagent (Invitrogen, cat. no.: 15596026, USA) and stored at −70 ˚C. We evaluated the quality and quantity of isolated RNA using Nanodrop (Thermo Scientific 2000, USA). Besides, cDNA was synthesized from 100 ng of isolated RNA using the RevertAid First Strand cDNA Synthesis Kit (ThermoFisher Scientific, cat. no.: K1622, USA). Finally, we performed quantitative real-time PCR (Applied Biosystems, USA) three times with cDNA replicates using SYBR Green PCR Master Mix (Applied Biosystems, USA). The utilized cycling conditions and thermal profile were as follows: 1 cycle at 95 ˚C for 10 min, 40 cycles at 95 ˚C for 10 s, 59–61 ˚C for 30 s (based on primers annealing), and melting curve analysis was performed ramping from 60 ˚C to 90 ˚C.

In this instance, we utilized the following primers which were designed by Gene Runner software:

CDH1 gene: forward: 5′GCTGTGTCATCCAACGGGAATG3′ and Reverse: 5′ GGGTGAATTCG GGCTTGTTGTC3′;

CDH2 gene: forward: 5′ GATAGCCCGGTTTCATTTGAGG3′ and Reverse: 5′ TGTCCCATTC CAAACCTGGTG3′;

MCP-1 gene: forward: 5′ GCATTGATTGCATCTGGCTG3′ and Reverse: 5′ TTCTCAAACTG AAGCTCGCAC3′;

PAI-1 gene: forward: 5′ TGAATTCCTGCAGCTCAGC3′ and Reverse: 5′ ACAGCAGACCCTTC ACCAAAG3′.

The relative quantification of CDH1, CDH2, MCP-1, and PAI-1 mRNAs expression was calculated by using $2^{-\Delta\Delta Ct}$ method [42]:

$$\Delta\Delta Ct = \left( Ct_{\text{target gene}} - Ct_{\text{Housekeeping gene}} \right)_{\text{patients}} - \left( Ct_{\text{target gene}} - Ct_{\text{Housekeeping gene}} \right)_{\text{control}}$$

Relative fold change of gene expression = $2^{-\Delta\Delta Ct}$

## Statistical analysis

All statistical analyses were conducted in SPSS 19.0 software. To compare study groups, a one-way analysis of variance (ANOVA) with Bonferroni post hoc test was executed in case of continuous normally distributed variables, nonparametric Kruskal-Wallis and Mann-Whitney tests were performed in case of continuous non-normally distributed variables, and a chi-square test was executed in case of categorical variables. Correlation of mRNA expression level with demographic covariates and clinical laboratory tests was assessed with the Spearman's correlation coefficient. Multivariable linear regression analysis using a stepwise method was performed to identify risk factors of mRNA expression. Receiver operating characteristic (ROC) analysis was used to explore the sensitivity, specificity, positive predictive value (PPV), negative predictive value (NPV), and area under the curve (AUC) of mRNA for predicting overt DN and incipient DN from DM. The best cutoff value for mRNA level was determined using Youden's J statistic. In the case that overt DN and incipient DN groups had lower mRNA expression levels than the DM group, the reverse fold change of expression level was considered for the ROC analysis. Continuous normally distributed variables were reported with mean and standard deviation, continuous non-normally distributed variables were reported with median and interquartile range, and categorical variables were reported with absolute number and percentage. $P$-value $< 0.05$ was considered as a statistically significant level in all tests.

## Results

### Demographic information and clinical laboratory characteristics

The demographic features and clinical laboratory test results of participants are depicted in Table 1. 196 participants with a mean age of $61.24 \pm 7.91$ years old were enrolled in this study. 119 (60.7%) of participants were male, and 47 had a positive family history of diabetes. Participants were divided into four groups, including 35 overt DN patients, 53 incipient DN patients, 62 DM patients, and 46 controls. These groups had no significant difference in age ($p = 0.980$), sex ($p = 0.137$), BMI ($p = 0.052$), and eGFR ($p = 0.265$). The study subjects had a mean BMI of $29.10 \pm 4.86$ kg/m$^2$ and a mean eGFR of $91.82 \pm 28.77$ ml/min/1.73m$^2$. The raw data of participants is available in S1 Data.

ANOVA analyses with multiple comparisons on clinical laboratory results (Table 2) showed that the overt DN group, incipient DN group, and DM group had higher FBS ($p \leq 0.001$) and HbA1c ($p < 0.001$) than controls. Also, overt DN group had higher FBS ($p = 0.022$) and HbA1c ($p = 0.002$) than DM group. Pairwise comparisons of overt DN group with incipient DN group and incipient DN group with DM group showed no significant difference for FBS ($p = 0.799$ and $p = 0.695$, respectively) and HbA1c ($p = 0.065$ and $p = 1.000$, respectively). Overt DN group had higher urea than DM group and control ($p < 0.001$ for both). Also, the incipient DN group had higher urea than the control ($p < 0.001$). However, no significant difference in urea was in pairwise comparisons of overt DN group with incipient DN group ($p = 0.422$), incipient DN group with DM group ($p = 0.081$), and DM group with control ($p = 0.312$). The concentration of creatinine in overt DN group was higher than in incipient

**Table 1. Demographic and clinical laboratory characteristics.**

| Variables | Total (n = 196) | Group | | | | P value |
|---|---|---|---|---|---|---|
| | | Overt DN (n = 35) | Incipient DN (n = 53) | DM (n = 62) | Control (n = 46) | |
| Age (years) | 61.24 ± 7.91 | 61.66 ± 8.88 | 61.41 ± 9.87 | 60.84 ± 6.86 | 61.38 ± 6.86 | 0.980 |
| Sex (M/F) (Male, %) | 119/77 (60.7%) | 27/8 (77.1%) | 31/22 (58.5%) | 33/29 (53.2%) | 28/18 (60.9%) | 0.137 |
| Family History (n, %) | 47 (24%) | 4 (11.4%) | 13 (24.5%) | 23 (37.1%) | 7 (15.2%) | 0.013 |
| Body Mass Index (kg/m$^2$) | 29.10 ± 4.86 | 26.92 ± 3.93 | 31.19 ± 4.86 | 29.72 ± 5.60 | 27.56 ± 3.28 | 0.052 |
| Fasting glucose (mg/dL) | 147.47 ± 69.79 | 186.26 ± 86.62 | 165.25 ± 87.67 | 146.50 ± 42.09 | 98.58 ± 14.41 | <0.001 |
| HbA1c (%) | 7.86 ± 1.97 | 9.26 ± 2.34 | 8.36 ± 1.60 | 8.01 ± 1.61 | 5.85 ± 0.45 | <0.001 |
| Urea (mg/dL) | 35.26 ± 14.90 | 44.56 ± 18.66 | 39.06 ± 18.85 | 32.65 ± 8.11 | 27.38 ± 6.67 | <0.001 |
| Creatinine (mg/dL) | 1.10 ± 0.37 | 1.38 ± 0.50 | 1.18 ± 0.41 | 0.92 ± 0.24 | 1.02 ± 0.16 | <0.001 |
| Urine Alb/Cr (mg/gr) | 181.38 ± 370.42 | 738.45 ± 587.66 | 150.61 ± 110.72 | 13.55 ± 12.14 | 8.13 ± 10.66 | <0.001 |
| eGFR (ml/min/1.73m2) | 91.82 ± 28.77 | 91.10 ± 41.68 | 94.74 ± 31.24 | 97.00 ± 28.90 | 83.47 ± 24.24 | 0.265 |

Continuous variables are reported with Mean ± SD and categorical variables are reported with number and percentage (n, %).

DN: diabetic nephropathy, DM: diabetes mellitus, HbA1c: glycated hemoglobin, Alb/Cr: albumin/ creatinine ratio, eGFR: estimated glomerular filtration rate

DN patients ($p = 0.044$), DM patients ($p < 0.001$), and control ($p < 0.001$). Incipient DN group had a higher creatinine concentration than DM group ($p = 0.001$) while incipient DN group had no significant difference in creatinine concentration compared to control ($p = 0.151$). In addition, there was no significant difference in creatinine concentration between the DM group and the control. ($p = 0.875$) As expected, the overt DN group and incipient DN group had higher urine Alb/Cr ratio than the DM group and control ($p \leq 0.040$). Also, the overt DN group had higher urine Alb/Cr ratio than the incipient DN group ($p < 0.001$), and no significant difference was in the urine Alb/Cr ratio between the DM group and control ($p = 1.000$).

## Characterization of blood EVs

Fig 1 shows the results of the size and morphology of isolated EVs analyzed by SEM and DLS. The results revealed that the size of isolated EVs was between 80–152 nm.

## mRNA expression level in blood EVs

The relative fold changes of CDH1, CDH2, MCP-1, and PAI-1 mRNAs in blood EVs of overt DN group, incipient DN group, and DM group compared to control are listed in Table 3. Kruskal–Wallis analyses showed no significant difference in CDH1 mRNA expression fold

**Table 2. Pairwise comparisons regarding clinical laboratory tests of overt DN group, incipient DN group, DM group, and control group versus each other.**

| Variables | Overt DN vs Incipient DN | Overt DN vs DM | Overt DN vs Control | Incipient DN vs DM | Incipient DN vs Control | DM vs Control |
|---|---|---|---|---|---|---|
| FBS | 0.799 | **0.022** | **<0.001** | 0.695 | **<0.001** | 0.001 |
| HbA1c | 0.065 | **0.002** | **<0.001** | 1.000 | **<0.001** | **<0.001** |
| Urea | 0.422 | **<0.001** | **<0.001** | 0.081 | **<0.001** | 0.312 |
| Creatinine | **0.044** | **<0.001** | **<0.001** | **0.001** | 0.151 | 0.875 |
| Alb/Cr | **<0.001** | **<0.001** | **<0.001** | **0.030** | **0.040** | 1.000 |

$p$ values are calculated by the Bonferroni post hoc test. FBS: fasting blood sugar, Alb/Cr: urine albumin/ creatinine ratio

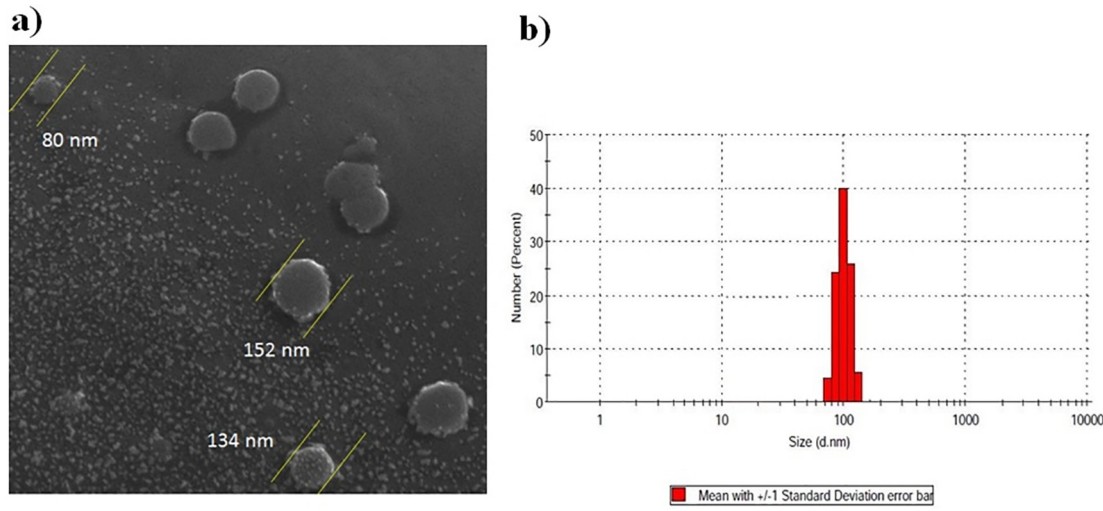

**Fig 1. SEM (a) and DLS (b) images of isolated EVs show the morphology and size of EVs.**

change between four study groups ($p$ = 0.408) while CDH2, MCP-1, and PAI-1 mRNAs had different expression levels between groups. ($p$< 0.001, $p$< 0.001, and $p$ = 0.017, respectively).

The fold changes of CDH2 mRNA in different groups are depicted in Fig 2. CDH2 mRNA was expressed less in blood EVs of overt DN group than incipient DN group ($p$ = 0.013), DM group ($p$< 0.001), and control ($p$< 0.001). Besides, the incipient DN group had lower CDH2 mRNA expression than the DM group ($p$ = 0.042). However, there were no significant differences in CDH2 mRNA level in pairwise comparisons of incipient DN group with control ($p$ = 0.061) and DM group with control ($p$ = 0.234).

Fig 3 shows MCP-1 mRNA expression fold change in different groups. As can be seen, the overt DN group had a lower MCP-1 mRNA level than the DM group and control ($p$ = 0.002 for both), while there was no significant difference in MCP-1 mRNA between the overt DN group and incipient DN group ($p$ = 0.262). Incipient DN group also expressed less MCP-1 mRNA compared to DM group ($p$ = 0.008) and control ($p$ = 0.001), and DM group expressed more MCP-1 mRNA than control ($p$ = 0.017). The pairwise comparisons of CDH1, CDH2, MCP-1, PAI-1 mRNAs between each of two groups of overt DN group, incipient DN group, DM group, and control group are summarized in Table 4.

**Table 3. Relative mRNA expression fold change in overt DN group, incipient DN group, and DM group compared to control.**

| mRNAs | Fold change* | | | Total P value** |
|---|---|---|---|---|
| | **Overt DN** | **Incipient DN** | **DM** | |
| **CDH1** | 0.79 (0.36, 1.35) | 0.87 (0.31, 1.59) | 0.71 (0.21, 1.98) | 0.408 |
| **CDH2** | 0.22 (0.10, 1.61) | 0.60 (0.19, 3.88) | 1.72 (0.35, 4.75) | <0.001 |
| **MCP-1** | 0.15 (0.07, 3.21) | 0.43 (0.06, 4.40) | 2.77 (0.14, 16.28) | <0.001 |
| **PAI-1** | 0.79 (0.41, 1.62) | 0.70 (0.34, 1.72) | 0.58 (0.23, 1.58) | 0.017 |

Data are expressed as median (IQR).

*Fold change was calculated by using $2^{-\Delta\Delta CT}$ method.

**Total $p$ value is calculated by the Kruskal–Wallis test.

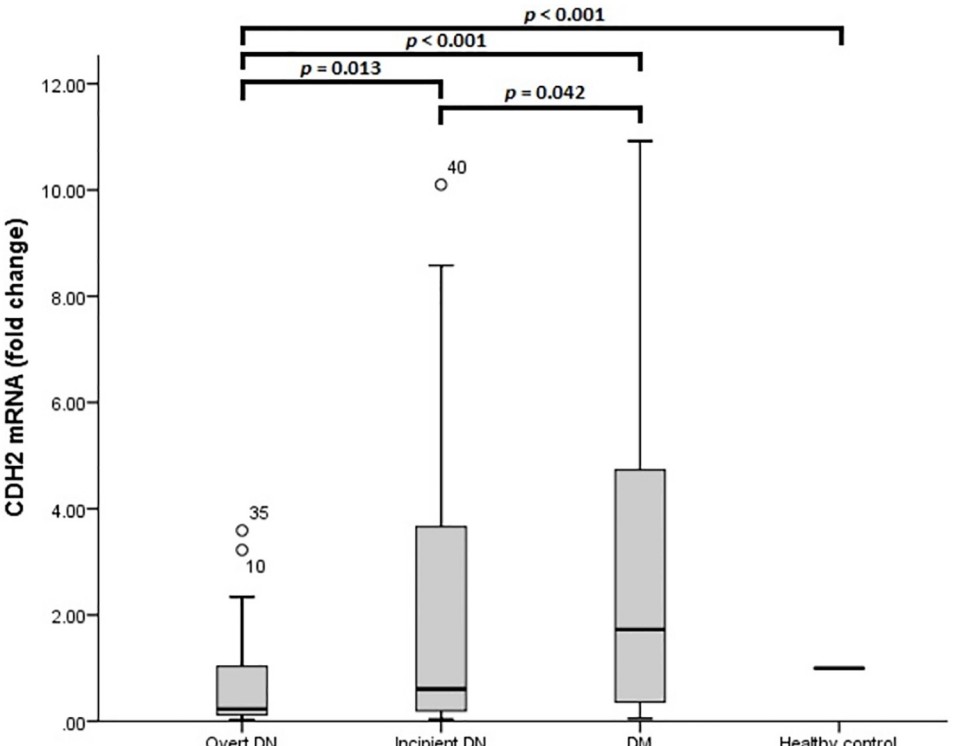

**Fig 2. CDH2 mRNA expression level in blood extracellular vesicles.** *p* value is calculated by the Mann–Whitney U test.

Pairwise comparisons in PAI-1 mRNA level show lower expression of PAI-1 mRNA in incipient DN group ($p = 0.011$) and DM group ($p = 0.003$) compared to control, while there were no significant differences in PAI-1 mRNA expression level between overt DN group and incipient DN group ($p = 0.963$), overt DN group and DM group ($p = 0.184$), overt DN group and control ($p = 0.089$), and incipient DN group and DM group ($p = 0.219$).

## Correlation between mRNA level and clinical laboratory tests

The correlations of CDH1, CDH2, MCP-1, and PAI-1 mRNAs with demographic features and clinical laboratory results are summarized in Table 5. Within all participants, CDH2 mRNA level inversely correlated with HbA1c (Spearman correlation coefficient = -0.183, $p = 0.012$), urea (Spearman correlation coefficient = -0.152, $p = 0.035$), creatinine (Spearman correlation coefficient = -0.370, $p < 0.001$), and urine Alb/Cr ratio (Spearman correlation coefficient = -0.355, $p < 0.001$). Besides, MCP-1 mRNA had negative correlations with creatinine (Spearman correlation coefficient = -0.361, $p < 0.001$) and urine Alb/Cr ratio (Spearman correlation coefficient = -0.297, $p < 0.001$).

Multivariate linear regression results on the parameters affecting CDH2 and MCP-1 mRNAs expression are depicted in Table 6. These analyses indicated that the expression level of CDH2 mRNA could be estimated from creatinine (unstandardized coefficient = -2.673, $p < 0.001$) and urea (unstandardized coefficient = 0.035, $p = 0.019$), and MCP-1 mRNA expression level could be estimated from creatinine level (unstandardized coefficient = -3.024, $p = 0.010$).

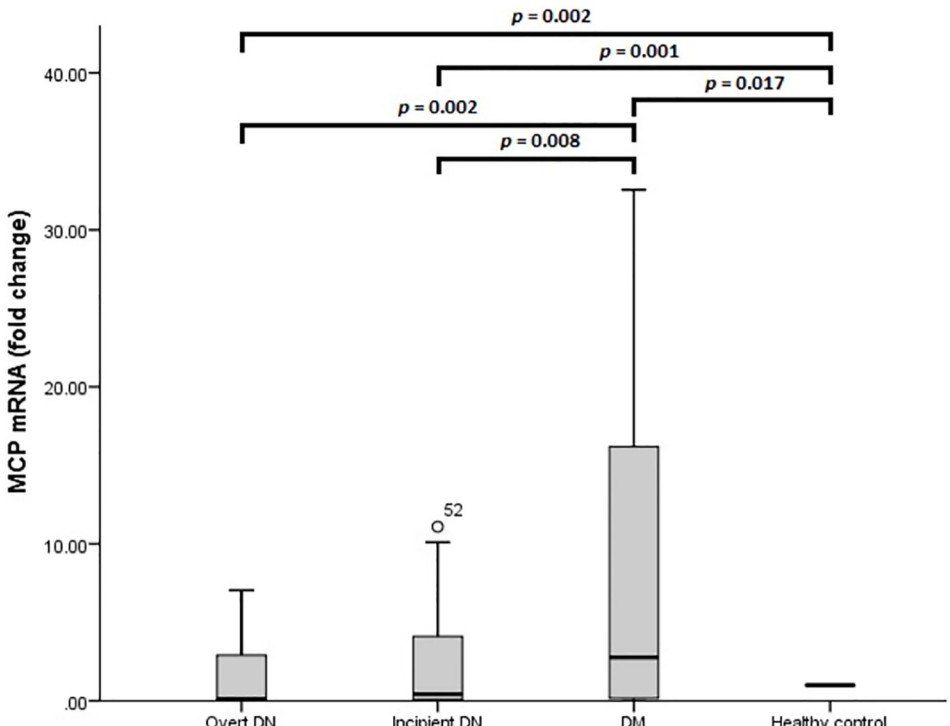

**Fig 3. MCP-1 mRNA expression level in blood extracellular vesicles.** *p* value is calculated by the Mann–Whitney U test.

## Diagnostic ability of 1/CDH2 and 1/MCP-1 mRNAs in incipient DN detection

The expression levels of CDH2 and MCP-1 mRNAs in the incipient DN group and overt DN group were lower than those in the DM group; thus, as described previously, ROC analysis was performed using the inverse fold changes of mRNA. Fig 4 shows the ROC curves of 1/CDH2 and 1/MCP-1 mRNAs for discrimination of incipient DN patients from diabetic patients. In this case, 1/CDH2 mRNA had an accuracy of 0.61 (95%CI: 0.50–0.71, *p* = 0.038), sensitivity of 37.7%, specificity of 83.9%, PPV of 66.6%, and NPV of 61.1% at a cutoff of 3.47 for incipient DN detection. In addition, the accuracy, sensitivity, specificity, PPV, and NPV of 1/MCP-1 mRNA ≥ 0.61 to detect incipient DN were 0.61 (95%CI: 0.51–0.71, *p* = 0.035), 69.8%, 61.3%, 60.6%, and 70.3%, respectively. Table 7 summarizes the diagnostic ability of 1/CDH2 and 1/MCP-1 mRNAs to distinguish incipient DN from DM.

**Table 4. Pairwise comparisons of overt DN group, incipient DN group, DM group, and control group versus each other.**

| mRNAs | Overt DN vs Incipient DN | Overt DN vs DM | Overt DN vs Control | Incipient DN vs DM | Incipient DN vs Control | DM vs Control |
|---|---|---|---|---|---|---|
| **CDH1** | 0.639 | 0.988 | 0.089 | 0.641 | 0.126 | 0.234 |
| **CDH2** | **0.013** | **<0.001** | **<0.001** | **0.042** | 0.061 | 0.234 |
| **MCP-1** | 0.262 | **0.002** | **0.002** | **0.008** | **0.001** | **0.017** |
| **PAI-1** | 0.963 | 0.184 | 0.089 | 0.219 | **0.011** | **0.003** |

*p* values are calculated by the Mann–Whitney U test.

**Table 5. Correlations of CDH1, CDH2, MCP-1, and PAI-1 mRNAs with demographic data and clinical laboratory tests in all participants.**

| | mRNA | | Age | BMI | FBS | HbA1c | Urea | Cr | eGFR | Alb/Cr |
|---|---|---|---|---|---|---|---|---|---|---|
| Total | CDH1 | r | -0.049 | -0.069 | 0.000 | -0.096 | -0.011 | 0.000 | -0.043 | -0.064 |
| | | p | 0.585 | 0.520 | 0.997 | 0.187 | 0.880 | 0.996 | 0.685 | 0.376 |
| | CDH2 | r | -0.008 | 0.050 | -0.093 | **-0.183** | **-0.152** | **-0.370** | 0.139 | **-0.355** |
| | | p | 0.930 | 0.640 | 0.195 | **0.012** | **0.035** | **<0.001** | 0.189 | **<0.001** |
| | MCP-1 | r | 0.015 | -0.081 | -0.028 | -0.111 | -0.044 | **-0.361** | 0.025 | **-0.297** |
| | | p | 0.867 | 0.445 | 0.697 | 0.129 | 0.541 | **<0.001** | 0.818 | **<0.001** |
| | PAI-1 | r | -0.083 | -0.069 | -0.068 | -0.086 | -0.011 | 0.038 | -0.043 | 0.008 |
| | | p | 0.359 | 0.518 | 0.343 | 0.236 | 0.876 | 0.596 | 0.687 | 0.908 |

BMI: body mass index, FBS: fasting blood sugar, Cr: serum creatinine, eGFR: estimated glomerular filtration rate, Alb/Cr: urine albumin/ creatinine ratio, r = Spearman correlation coefficient, p = *p*-value.

### Diagnostic ability of 1/CDH2 and 1/MCP-1 mRNAs in overt DN detection

ROC curves of 1/CDH2 and 1/MCP-1 mRNAs for discrimination of overt DN from DM are depicted in Fig 5. 1/CDH2 mRNA showed good diagnostic accuracy with an AUC of 0.75 (95%CI: 0.65–0.85, $p<0.001$). Based on Youden's J statistic, 1/CDH2 mRNA had optimal diagnostic performance at 2.14-fold change. At this threshold, 1/CDH2 mRNA had a sensitivity of 74.3%, specificity of 69.4%, PPV of 57.8%, and NPV of 82.7% for overt DN detection. Besides, the accuracy 1/MCP-1 mRNA for overt DN detection was 0.66 (95%CI: 0.55–0.77, $p = 0.007$), and 1/MCP-1 mRNA level $\geq 5.89$ as the best cutoff value resulted in 57.1% as sensitivity, 74.2% as specificity, 55.5% as PPV, and 75.3% as NPV for discrimination of overt DN from DM. The diagnostic ability of 1/CDH2 and 1/MCP-1 mRNAs to distinguish overt DN from DM is summarized in Table 8.

### Discussion

In this study, we assessed the expression levels of CDH1, CDH2, MCP-1, and PAI-1 mRNAs in blood EVs of overt DN, incipient DN, and DM patients compared to healthy controls and

**Table 6. Multivariable linear regression analysis on the parameters affecting CDH2 and MCP-1 mRNAs level.**

| | Dependent variable: CDH2 mRNA | | | | | |
|---|---|---|---|---|---|---|
| Predictors[1] | Unstandardized coefficients | | Standardized coefficients | | t | P-value* |
| | B | SE | Beta | | | |
| Constant | 3.525 | 0.509 | | | 6.926 | <0.001 |
| Cr | -2.673 | 0.592 | -0.429 | | -4.516 | <0.001 |
| Urea | 0.035 | 0.015 | 0.225 | | 2.367 | 0.019 |
| | Dependent variable: MCP-1 mRNA | | | | | |
| Predictors[2] | Unstandardized coefficients | | Standardized coefficients | | t | P-value* |
| | B | SE | Beta | | | |
| Constant | 6.884 | 1.347 | | | 5.110 | <0.001 |
| Cr | -3.024 | 1.158 | -0.186 | | -2.611 | 0.010 |

[1] HbA1c, urea, serum creatinine, and urine Alb/Cr ratio were adopted for stepwise multivariable linear regression analysis of CDH2 mRNA.

[2] Serum creatinine and urine Alb/Cr ratio were adopted for stepwise multivariable linear regression analysis of MCP-1 mRNA.

* Statistically significance was set at *p* value < 0.05.

Cr: serum creatinine, SE: standard error.

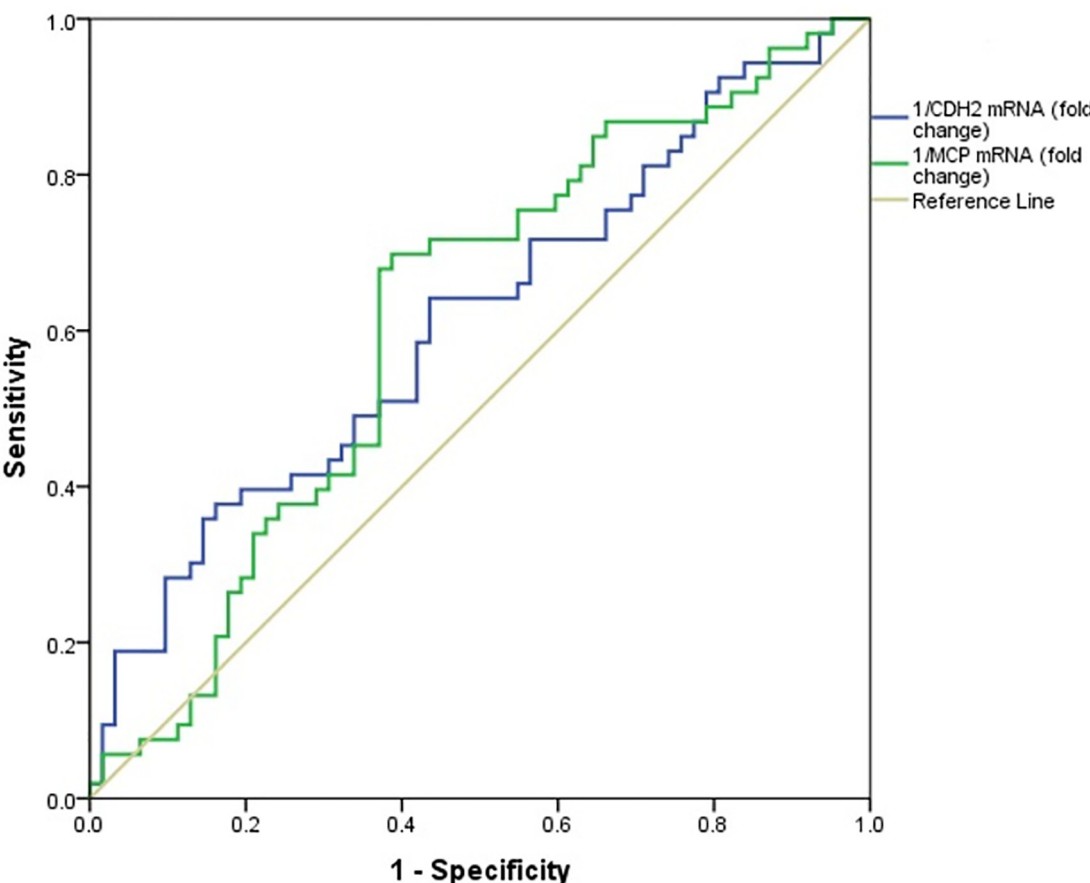

**Fig 4.  The receiver operating characteristic (ROC) analyses of 1/CDH2 mRNA and 1/MCP-1 mRNA to differentiate incipient DN patients from DM patients.**

investigated whether or not these mRNAs could serve as diagnostic biomarkers for early-stage DN. The results showed that CDH2 and MCP-1 mRNAs were downregulated in DN groups compared to the DM group, whereas the expression levels of CDH1 and PAI-1 mRNAs did not significantly differ between DN groups and the DM group. Moreover, CDH2 and MCP-1 mRNAs inversely correlated with the severity of albuminuria and showed excellent diagnostic ability to discriminate DN from DM. In this case, CDH2 and MCP-1 mRNAs from blood EVs not only have diagnostic potential for early-stage DN, but also could be promising target points to understand underlying mechanisms of DN progression and develop an alternative approach for the prevention, prognosis, and treatment of DN.

To date, several studies have been conducted to investigate the level of miRNAs and mRNAs derived from circulating EVs [43, 44] and urine EVs in DN [21, 22, 24, 45–48].

**Table 7.  Diagnostic performance of 1/CDH2 mRNA and 1/MCP-1 mRNA for incipient DN detection.**

| Parameters | Cutoff values | Sensitivity (%) | Specificity (%) | PPV (%) | NPV (%) | AUC (95%CI) | p value |
|---|---|---|---|---|---|---|---|
| 1/CDH2 | 3.47 | 37.7 | 83.9 | 66.6 | 61.1 | 0.61 (0.50, 0.71) | 0.038 |
| 1/MCP-1 | 0.61 | 69.8 | 61.3 | 60.6 | 70.3 | 0.61 (0.51, 0.71) | 0.035 |

PPV: positive predictive value, NPV: negative predictive value, AUC: area under the curve

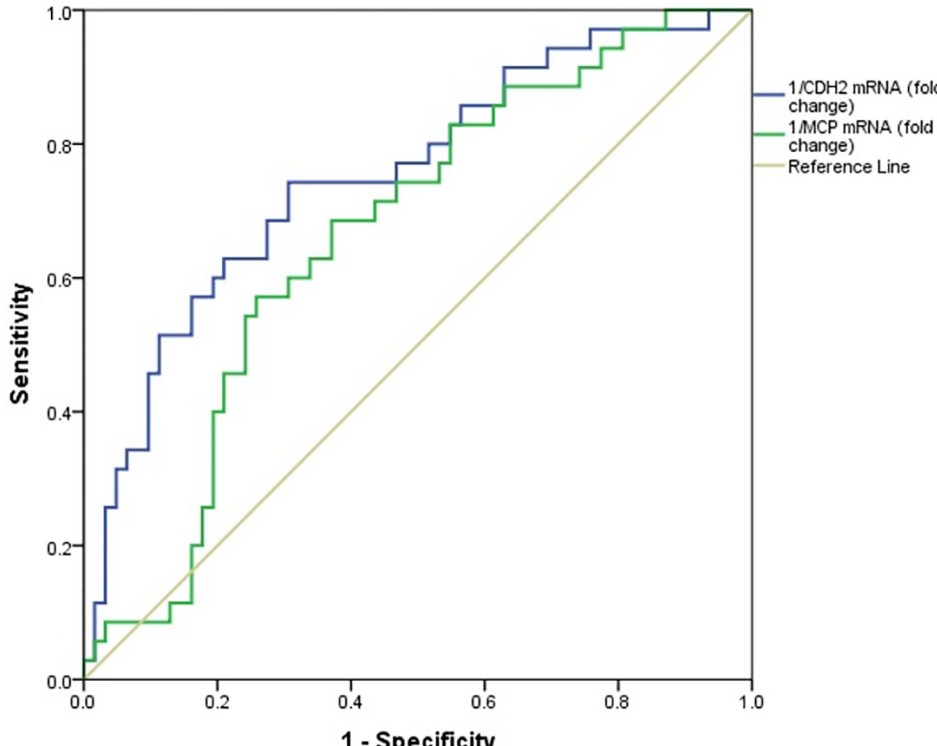

**Fig 5. The receiver operating characteristic (ROC) analyses of 1/CDH2 mRNA and 1/MCP-1 mRNA to differentiate overt DN patients from DM patients.**

However, to the best of our knowledge, this study is the first assessment of these specific mRNAs derived from blood EVs of DN patients. Previously, a nine patients-trial by Zheng M et al. [25] characterized the expression profile of CDH1, CDH2, MCP-1, and PAI-1 mRNAs in the urinary sediment of DN patients and reported significantly increased expression levels of these mRNAs in the DN group compared to healthy controls. They also found that CDH2 mRNA in the DN group had a 15-fold higher expression level than in control. Although this trial indicated upregulation of CDH1 and PAI-1 mRNAs of urinary pellet in DN, our study showed no significant change in CDH1 and PAI-1 mRNA expression in blood EVs of DN patients compared to DM patients and controls; meanwhile, PAI-1 mRNA was downregulated in DM patients compared to controls.

A multiple microarray analysis by Zhang Y et al. [39] revealed a positive correlation between CDH2 mRNA expression and eGFR in DN, suggesting a protective role of CDH2 mRNA expression in DN patients. Our analysis revealed no significant correlation between

**Table 8. Diagnostic performance of 1/CDH2 mRNA and 1/MCP-1 mRNA for overt DN detection.**

| Parameters | Cutoff values | Sensitivity (%) | Specificity (%) | PPV (%) | NPV (%) | AUC (95%CI) | p value |
|---|---|---|---|---|---|---|---|
| 1/CDH2 | 2.14 | 74.3 | 69.4 | 57.8 | 82.7 | 0.75 (0.65, 0.85) | <0.001 |
| 1/MCP-1 | 5.89 | 57.1 | 74.2 | 55.5 | 75.3 | 0.66 (0.55, 0.77) | 0.007 |

PPV: positive predictive value, NPV: negative predictive value, AUC: area under the curve

eGFR and CDH2 and MCP-1 mRNA expression. One reason for this finding is that four groups in our study had no significant difference in eGFR, and all participants had normal eGFR despite the different degrees of albuminuria. Thus, this result needs to be confirmed in a future study in which the participants have different levels of eGFR. In addition, because DN patients had normal eGFR, indicating good renal function, they were considered to be in the early stages of diabetic nephropathy. Therefore, we interpreted that the downregulation of CDH2 and MCP-1 mRNAs could be good predictors of early-stage DN.

In addition, several studies indicated the overexpression of urinary MCP-1 mRNA and protein in DN [28, 49–51] and showed that the expression of glomerular MCP-1 mRNA positively correlated with urinary protein excretion rate and kidney injuries [52]. Our analysis showed that CDH2 and MCP-1 mRNA expression had an inverse correlation with urine Alb/Cr ratio and serum creatinine, reflecting their renoprotective effect, and their downregulated expression correlated with impaired kidney. In addition, multiple regression revealed that serum creatinine was one of the predictors of CDH2 and MCP-1 mRNAs expression. However, this is an observational study, and we should interpret the results in terms of an associative model rather than a causal relationship. Thus, we require further experimental studies to determine the direction of associations between covariates.

All discussed studies had investigated the expression profile of CDH1, CDH2, MCP-1, and PAI-1 mRNAs in various biofluids other than blood EVs. However, recent quantitative RNA sequencing of WT1 and ACE mRNAs from blood EVs of DN patients showed upregulated expression of WT1 mRNA and downregulated expression of ACE mRNA in DN patients compared to DM patients and controls. Moreover, WT1 and ACE mRNAs from blood EVs were associated with the severity of albuminuria and showed good diagnostic performance to discriminate DN from DM [41]. This study, in agreement with our study, indicated the important role of mRNA from blood EVs in DN development and also showed that exploring more mRNA profiles of blood EVs in DN will lead to a better understanding of DN pathophysiology and the discovery of alternative biomarkers for DN management.

This study had several limitations that should be expained. First, we conducted a cross-sectional study, and we could not conclude any causal interpretations. Second, we attempted to recruit age- and sex-matched control cases. Nevertheless, the study groups differed significantly in other demographic features such as family history and clinical laboratory tests that might influence the mRNA expression profile of blood EVs. Third, we identified DN cases based on micro/macroalbuminuria in three spot urine samples, whereas the gold standard for DN diagnosis is renal biopsy. Fourth, we did not examine the medications that might differ between study groups and influence the mRNAs expression. Fifth, EVs isolated from blood most likely originate from different cell types, and EVs originating from different cells most likely carry different cargos. Finally, to confirm our findings, we need a multicenter study with a large sample size and diverse participants in ethnicity.

## Conclusions

To sum up, CDH2 and MCP-1 mRNAs of blood EVs were downregulated in DN patients and showed good diagnostic potential for early detection of DN. Besides, the expression levels of these biomarkers are inversely correlated with serum creatinine, degree of albuminuria, and renal injury. Our study suggests novel mediators involved in DN pathogenesis. Further research on mRNA profiling of blood EVs is warranted to uncover new molecular pathways in DN development and may be suggested for the diagnosis, prognosis, prevention, and therapy of DN.

## Supporting information

**S1 Data.**

(XLSX)

## Author Contributions

**Conceptualization:** Katayoon Forouzanfar, Samaneh Zeidi, Negar Salehi, Farideh Razi, Ehsan Hashemi.

**Data curation:** Hojat Dehghanbanadaki.

**Formal analysis:** Hojat Dehghanbanadaki, Farideh Razi.

**Investigation:** Ardeshir Kakaei, Ehsan Hashemi.

**Methodology:** Hojat Dehghanbanadaki, Ardeshir Kakaei, Samaneh Zeidi, Ehsan Hashemi.

**Project administration:** Babak Arjmand.

**Resources:** Katayoon Forouzanfar, Babak Arjmand.

**Supervision:** Babak Arjmand, Farideh Razi.

**Validation:** Farideh Razi.

**Visualization:** Negar Salehi, Farideh Razi.

**Writing – original draft:** Hojat Dehghanbanadaki, Farideh Razi, Ehsan Hashemi.

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
