## [Decision Letter · Decision Letter 0]

26 Oct 2021

PONE-D-21-29236The Role of CDH1, CDH2,MCP, and PAI mRNAs of Blood Extracellular Vesicles in PredictingEarly-stage Diabetic NephropathyPLOS ONE

Dear Dr. Hashemi,

Thank you for submitting your manuscript to PLOS ONE. After careful consideration, we feel that it has merit but does not fully meet PLOS ONE’s publication criteria as it currently stands. Therefore, we invite you to submit a revised version of the manuscript that addresses the points raised during the review process.

We look forward to receiving your revised manuscript.

Kind regards,

Muhammad Tarek Abdel Ghafar, M.D

Academic Editor

PLOS ONE

Journal Requirements:

2. Please include your tables as part of your main manuscript and remove the individual files. Please note that supplementary tables should be uploaded) as separate "supporting information" files.

 [The funders had no role in study design, data collection and analysis, decision to publish, or preparation of the manuscript.]. 

5. PLOS requires an ORCID iD for the corresponding author in Editorial Manager on papers submitted after December 6th, 2016. Please ensure that you have an ORCID iD and that it is validated in Editorial Manager. To do this, go to ‘Update my Information’ (in the upper left-hand corner of the main menu), and click on the Fetch/Validate link next to the ORCID field. This will take you to the ORCID site and allow you to create a new iD or authenticate a pre-existing iD in Editorial Manager. Please see the following video for instructions on linking an ORCID iD to your Editorial Manager account: https://www.youtube.com/watch?v=_xcclfuvtxQ.

Reviewers' comments:

Reviewer's Responses to Questions

**Comments to the Author**

1. Is the manuscript technically sound, and do the data support the conclusions?

Reviewer #1: Partly

Reviewer #2: Yes

2. Has the statistical analysis been performed appropriately and rigorously? 

Reviewer #1: Yes

Reviewer #2: Yes

3. Have the authors made all data underlying the findings in their manuscript fully available?

Reviewer #1: Yes

Reviewer #2: Yes

4. Is the manuscript presented in an intelligible fashion and written in standard English?

Reviewer #1: No

Reviewer #2: No

5. Review Comments to the Author

Reviewer #1: Reviewer Response:

Following amendments and corrections may further improve this manuscript.

1) The manuscript needs a thorough revision regarding its “space” formatting, in all the sections including “Title-statement”.

2) Kindly review, Line and Paragraph-spacing.

3) The manuscript needs a thorough revision for its Grammarly- mistakes; use of helping verbs and tenses.

4) Kindly, add Line numbers, as per journal’s format.

5) Section: Abstract: Methods: Line#4: Kindly abbreviate ROC.

6) Section: Abstract: Results: Line#1: Kindly recorrect the statement as; “The mRNA expression found in CDH2 and MCP, was down-regulated in overt DN group”.

7) Section: Abstract: Results: Line#1: Kindly recorrect as: “0.22- and 0.15-fold change” to “0.22- fold change and 0.15-fold change”, respectively.

8) Section: Abstract: Results: Line#2: Kindly recorrect as: “0.60- and 0.43-fold change” to “0.60- fold change and 0.43-fold change”.

9) Section: Abstract: Results: Line#3: Kindly recorrect as: (1.72- and 2.77-fold change) to (1.72-fold change and 2.77-fold change).

10) Section: Abstract: Results: Line#4: Kindly recorrect as: DM group (0.58-fold change) as compare to control.

11) Section: Abstract: Results: Line#8: (95%CI: 0.65, 0.85) should be written as (95%CI: 0.65-0.85), vice versa throughout the text.

12) Section: Introduction:1st Paragraph: Line 22/23: “while we now know EVs”, should be corrected as “While, now we know that EVs”.

13) It is recommended to correct “mRNAs” throughout the text as “mRNA”, in properly manner.

14) Section: Methods: Subjects and ethical considerations: Line 7 & 9: Kindly correct “creatinineon” as “creatinine on”.

15) Section: Methods: mRNA isolation, cDNA preparation and qRT-PCR analysis: Kindly, add thermo-cycle programming in Tabular-Form;

Initial Denaturation, Denaturation, Annealing, Extension and Final Extension: timing and temperature/ cycle.

16) Section: Results: Demographic information and clinical laboratory characteristics: Line#20: “significant different creatinineconcentration” should be corrected as “significant difference in creatinine concentration”.

17) Section: Results: Diagnostic ability of 1/CDH2 and 1/MCP mRNAs in overt DN detection: Line#1: Kindly, add helping verb as: “discrimination of overt DN from DM”.

18) Section: Results: Diagnostic ability of 1/CDH2 and 1/MCP mRNAs in overt DN detection: Line#3: What do you mean by guaranteed sensitivity=?

19) Section: Results: Diagnostic ability of 1/CDH2 and 1/MCP mRNAs in overt DN detection: Line# 3rd last Line: should be corrected as: ‘for discrimination of overt DN from DM.

20) Section: Discussion: 1st Para: 2nd last Line: “DN progression and develop”, should be corrected as: “DN progression and development”.

21) The manuscript doesn’t include the brief introduction of CDH1, CDH2, MCP & PAI, neither their abbreviation nor their function. No information or Literature Review regarding: How they act as Biomarkers for disease detection, particularly DN? How they help to investigate DN manifestation? Either, they are genes or proteins or their molecular pathway is associated with DN related Risk Factors? Need to Elaborate exclusively.

22) Kindly also provide with brief explanation of Biomarkers and how they help to investigate disease progression?

Reviewer #2: Comments

The authors addressed diabetic nephropathy (DN) patients in their study. They studied the expression of a panel of DN related genes in blood extracellular vesicles (EVs) of 4 groups; overt DN patients, incipient DN patients, diabetes mellitus patients (DM), and healthy controls. They reported that CDH2 and MCP mRNA expression is significantly downregulated in blood EVs of overt and incipient DN patients relative to DM patients lacking DN. CDH2 gene was found more downregulated in EVs of overt DN patients compared to incipient and DM patients. Expression of both genes was inversely correlated with serum creatinine levels and degree of albuminuria. They suggested that CDH2 and MCP mRNA expression declines as DN develops, indicating a possible renoprotective effect of these mRNAs in diabetic individuals and their possible role as diagnostic biomarkers for early-stage DN that may allow monitoring its progression. The statistical studies are very good. The study is interesting and concerns a considerable number of researchers and physicians.

1) The full names of the investigated genes should be written in the introduction with a referral to their functions.

2) In the methods section, Subjects and ethical considerations sub-section, please, preferably mention the type of diabetes mellitus from which the samples were obtained. Which type of DM was addressed? or were both types included in the study?

3) In the methods section, Sample collection and clinical laboratory tests section, please, put a reference for the Cockcroft–Gault Equation used in this section.

4) In the methods section, mRNA isolation, cDNA preparation and qRT-PCR analysis sub-section:

- “TRIzolTM Reagent(USA, Cat. Num: 15596026. Germany)”: please, mention the company and revise the manufacturer’s country.

-“ RevertAid First Strand cDNA Synthesis Kit (Fermentas, Cat Num: K1622, USA), Is it Fermentas or Thermo Scientific?

- “Nanodrop (Thermo Scientific 2000)”- “SYBR Green PCR Master Mix (Applied Biosystems)”: please mention the country.

- Were the used primers designed by authors or obtained from previous studies? , if so, please, mention the reference(s).

5) In the Results section, Demographic information and clinical laboratory characteristics subsection:

- In table 1, does the p value represent the difference between the 4 groups? It will be better to put the p value of the pairwise comparison (the difference between each 2 groups) by subdividing the p value column so that the results become clearer.

- For the creatinin concentration results, in the following part: “Incipient DN group had higher creatinine concentration than DM group (p = 0.001) and no significant different creatinine concentration compared to control (p = 0.151).”

Do you mean no significant difference between DN group and control group or between DM group and control group? Please, clarify.

6) In the Results section, Characterization of blood Evs subsection, in the following part:

“Figure 1 showed the results of the size and morphology of isolated exosomes which were analyzed by SEM and DLS. The results revealed that size of isolated EVs were between 80-150 nm”, you mentioned that the isolated EVs are exosomes, however the size stated was “80-150 nm” which means that not all the isolated EVs are exosomes according to the classification mentioned in the introduction: “Exosomes are 30–100 nm in diameter, derived from the endosomal pathway and fusion of multivesicular bodies to plasma membranes, while microvesicles, with a diameter of 100-1,000 nm, are derived from budding and evagination of plasma membranes.(16)”

-Figure 1a shows an EV with a diameter of 152 nm so the size range would be: (80-152 nm) to be more precise.

7) In the Results section, mRNAs expression level in blood EVs subsection:

-In table 2, it is better to present the p values of the pairwise comparison (presented later in Table S 1 with no referral to this table in the text), as these results are important and should be presented in the main table (Table 2) not in a supplementary one.

6. PLOS authors have the option to publish the peer review history of their article (what does this mean?). If published, this will include your full peer review and any attached files.

Reviewer #1: **Yes: **Syeda Zahra Abbas

Reviewer #2: No

---

## [Author Response · Author response to Decision Letter 0]

26 Dec 2021

Response to Reviewers

Reviewer #1: Reviewer Response:

Following amendments and corrections may further improve this manuscript.

1) The manuscript needs a thorough revision regarding its “space” formatting, in all the sections including “Title-statement”.

Done, you are right. Thanks for your appropriate comment. We correct “space” formatting of all part of the text in the revised manuscript.

2) Kindly review, Line and Paragraph-spacing.

Done, Thanks for your comment. Manuscript text was double-spaced.

3) The manuscript needs a thorough revision for its Grammarly- mistakes; use of helping verbs and tenses.

Done, Thanks for your precise comment. We corrected the Grammarly- mistakes to the best of our knowledge.

4) Kindly, add Line numbers, as per journal’s format.

Done, you are right. We added continuous Line numbers.

5) Section: Abstract: Methods: Line#4: Kindly abbreviate ROC.

Done, you are right. We wrote the full name of ROC in this part of revised manuscript.

6) Section: Abstract: Results: Line#1: Kindly recorrect the statement as; “The mRNA expression found in CDH2 and MCP, was down-regulated in overt DN group”.

Done, Thanks for your excellent comment. We recorrect this statement in the abstract section of revised manuscript.

7) Section: Abstract: Results: Line#1: Kindly recorrect as: “0.22- and 0.15-fold change” to “0.22- fold change and 0.15-fold change”, respectively.

Done, Thanks for your precise comment. We recorrected this statement in the revised manuscript.

8) Section: Abstract: Results: Line#2: Kindly recorrect as: “0.60- and 0.43-fold change” to “0.60- fold change and 0.43-fold change”.

Done, Thanks for your precise comment. We recorrected this statement in the revised manuscript.

9) Section: Abstract: Results: Line#3: Kindly recorrect as: (1.72- and 2.77-fold change) to (1.72-fold change and 2.77-fold change).

Done, Thanks for your comment. We recorrected this statement in the revised manuscript.

10) Section: Abstract: Results: Line#4: Kindly recorrect as: DM group (0.58-fold change) as compare to control.

Done, Thanks for your appropriate comment. We recorrected it in the revised manuscript.

11) Section: Abstract: Results: Line#8: (95%CI: 0.65, 0.85) should be written as (95%CI: 0.65-0.85), vice versa throughout the text.

Done, Thanks for your promising comment. We recorrect this issue throughout the text.

12) Section: Introduction:1st Paragraph: Line 22/23: “while we now know EVs”, should be corrected as “While, now we know that EVs”.

Done, Thanks a lot for your precise comment. We recorrect this statement.

13) It is recommended to correct “mRNAs” throughout the text as “mRNA”, in properly manner.

Done, thanks for your promising comment. We correct “mRNAs” as “mRNA” throughout the text in proper manner.

14) Section: Methods: Subjects and ethical considerations: Line 7 & 9: Kindly correct “creatinineon” as “creatinine on”.

Done, you are right. We correct it. Besides, we revised the text in the manner of “spacing”.

15) Section: Methods: mRNA isolation, cDNA preparation and qRT-PCR analysis: Kindly, add thermo-cycle programming in Tabular-Form;

Initial Denaturation, Denaturation, Annealing, Extension and Final Extension: timing and temperature/ cycle.

Done, Thanks for your comment.

16) Section: Results: Demographic information and clinical laboratory characteristics: Line#20: “significant different creatinineconcentration” should be corrected as “significant difference in creatinine concentration”.

Done, Thanks for your comment. We correct this.

17) Section: Results: Diagnostic ability of 1/CDH2 and 1/MCP mRNAs in overt DN detection: Line#1: Kindly, add helping verb as: “discrimination of overt DN from DM”.

Done, Thanks for your precise comment. We corrected it in the revised manuscript.

18) Section: Results: Diagnostic ability of 1/CDH2 and 1/MCP mRNAs in overt DN detection: Line#3: What do you mean by guaranteed sensitivity=?

Done, you are right. This sentence was vague; so we replaced it with “Based on Youden's J statistic, 1/CDH2 mRNA had optimal diagnostic performance at 2.14-fold change. In this cutoff, 1/CDH2 mRNA had sensitivity of 74.3%, specificity of 69.4%, PPV of 57.8%, and NPV of 82.7% for overt DN detection.”. In this sentence, we reported the sensitivity, specificity, PPV, and NPV of 1/CDH2 mRNA at cutoff of 2.14-fold change.

19) Section: Results: Diagnostic ability of 1/CDH2 and 1/MCP mRNAs in overt DN detection: Line# 3rd last Line: should be corrected as: ‘for discrimination of overt DN from DM.

Done, Thanks for your precise comment. We corrected it in the revised manuscript.

20) Section: Discussion: 1st Para: 2nd last Line: “DN progression and develop”, should be corrected as: “DN progression and development”.

Done, Thanks for your precise comment. We corrected it.

21) The manuscript doesn’t include the brief introduction of CDH1, CDH2, MCP & PAI, neither their abbreviation nor their function. No information or Literature Review regarding: How they act as Biomarkers for disease detection, particularly DN? How they help to investigate DN manifestation? Either, they are genes or proteins or their molecular pathway is associated with DN related Risk Factors? Need to Elaborate exclusively.

Done, Thanks for your precise comment. We wrote the full name of the investigated genes and their abbreviations. We explained the productions and function of the investigated genes. We explained the pathological conditions related to the abnormal expression of the investigated genes. We provided the literature review regarding the potential ability of the investigated mRNAs (the transcript of investigated genes) that act as Biomarkers for renal fibrosis detection, especially due to DN. We wrote the usefulness of the investigated mRNAs for DN manifestations including “these biomarkers can accurately detect DN patients in its early-stage, stratify them into different stages for individualized management, and monitor their response to therapy”. We presented that the expression mRNA levels of the investigated genes were evaluated to identify whether they were associated with DN or not.

Introduction section; page 3; lines 136-139

Introduction section; page 4; lines 164-170

Introduction section; page 5; lines 170-190

22) Kindly also provide with brief explanation of Biomarkers and how they help to investigate disease progression?

Done, Thanks for your excellent comment. We explained the usefulness of new biomarkers, especially when combined with currently used biomarkers, for detecting, stratifying, and monitoring disease progression in the introduction section of revised manuscript. Introduction section; page 3; lines 136-139

Reviewer #2: Comments

The authors addressed diabetic nephropathy (DN) patients in their study. They studied the expression of a panel of DN related genes in blood extracellular vesicles (EVs) of 4 groups; overt DN patients, incipient DN patients, diabetes mellitus patients (DM), and healthy controls. They reported that CDH2 and MCP mRNA expression is significantly downregulated in blood EVs of overt and incipient DN patients relative to DM patients lacking DN. CDH2 gene was found more downregulated in EVs of overt DN patients compared to incipient and DM patients. Expression of both genes was inversely correlated with serum creatinine levels and degree of albuminuria. They suggested that CDH2 and MCP-1 mRNA expression declines as DN develops, indicating a possible renoprotective effect of these mRNAs in diabetic individuals and their possible role as diagnostic biomarkers for early-stage DN that may allow monitoring its progression. The statistical studies are very good. The study is interesting and concerns a considerable number of researchers and physicians.

1) The full names of the investigated genes should be written in the introduction with a referral to their functions.

Done, you are right. Thanks for your appropriate comment. We added the full names of the investigated genes as well as their functions in the introduction section of the revised manuscript.

Introduction section; page 4; lines 164-170

Introduction section; page 5; lines 170-188

2) In the methods section, Subjects and ethical considerations sub-section, please, preferably mention the type of diabetes mellitus from which the samples were obtained. Which type of DM was addressed? or were both types included in the study?

Done, Thanks for your suitable comment. All participants had type 2 diabetes mellitus and we mentioned it in the method section of revised manuscript.

3) In the methods section, Sample collection and clinical laboratory tests section, please, put a reference for the Cockcroft–Gault Equation used in this section.

Done, Thanks for your comment. We added the reference for Cockcroft–Gault Equation in the revised manuscript.

4) In the methods section, mRNA isolation, cDNA preparation and qRT-PCR analysis sub-section:

- “TRIzolTM Reagent(USA, Cat. Num: 15596026. Germany)”: please, mention the company and revise the manufacturer’s country.

-“ RevertAid First Strand cDNA Synthesis Kit (Fermentas, Cat Num: K1622, USA), Is it Fermentas or Thermo Scientific?

- “Nanodrop (Thermo Scientific 2000)”- “SYBR Green PCR Master Mix (Applied Biosystems)”: please mention the country.

- Were the used primers designed by authors or obtained from previous studies? , if so, please, mention the reference(s).

Done, Thanks for your comment.

5) In the Results section, Demographic information and clinical laboratory characteristics subsection:

- In table 1, does the p value represent the difference between the 4 groups? 

Yes, you are right. This p value represents the difference between the 4 groups.

It will be better to put the p value of the pairwise comparison (the difference between each 2 groups) by subdividing the p value column so that the results become clearer.

Done, Thanks for your appropriate comment. We did the pairwise comparisons of parameters that were significantly different between the 4 groups in Table 1 and we showed the pairwise comparisons in Table 2.

- For the creatinin concentration results, in the following part: “Incipient DN group had higher creatinine concentration than DM group (p = 0.001) and no significant different creatinine concentration compared to control (p = 0.151).”

Do you mean no significant difference between DN group and control group or between DM group and control group? Please, clarify.

Done, you are right. This statement was vague. We mean no significant difference between incipient DN group and control group. We clarified this finding in the revised manuscript.

6) In the Results section, Characterization of blood Evs subsection, in the following part:

“Figure 1 showed the results of the size and morphology of isolated exosomes which were analyzed by SEM and DLS. The results revealed that size of isolated EVs were between 80-150 nm”, you mentioned that the isolated EVs are exosomes, however the size stated was “80-150 nm” which means that not all the isolated EVs are exosomes according to the classification mentioned in the introduction: “Exosomes are 30–100 nm in diameter, derived from the endosomal pathway and fusion of multivesicular bodies to plasma membranes, while microvesicles, with a diameter of 100-1,000 nm, are derived from budding and evagination of plasma membranes.(16)”

Done, you are absolutely right. The isolated EVs were both exosomes and microvesicles. Therefore, we revised the manuscript and in every statement that we wrongly stated exosomes, we replaced them with extracellular vesicle (EV).

-Figure 1a shows an EV with a diameter of 152 nm so the size range would be: (80-152 nm) to be more precise.

Done, you are right. We corrected it in the revised manuscript.

7) In the Results section, mRNAs expression level in blood EVs subsection:

-In table 2, it is better to present the p values of the pairwise comparison (presented later in Table S 1 with no referral to this table in the text), as these results are important and should be presented in the main table (Table 2) not in a supplementary one.

Done, you are right. Thanks for your appropriate comment. We presented Table S1 as Table 4 in the main text.

---

## [Decision Letter · Decision Letter 1]

19 Jan 2022

PONE-D-21-29236R1The role of CDH1, CDH2, MCP-1, and PAI-1 mRNAs of blood extracellular vesicles in predicting early-stage diabetic nephropathyPLOS ONE

Dear Dr. Hashemi,

Thank you for submitting your manuscript to PLOS ONE. After careful consideration, we feel that it has merit but does not fully meet PLOS ONE’s publication criteria as it currently stands. Therefore, we invite you to submit a revised version of the manuscript that addresses the points raised during the review process.

We look forward to receiving your revised manuscript.

Kind regards,

Muhammad Tarek Abdel Ghafar, M.D

Academic Editor

PLOS ONE

Journal Requirements:

Additional Editor Comments (if provided):

There are several structural, grammatical, punctuation, and capitalization errors that require heavy editing by a native English speaker. Please revise the organization of the tables as there are supplementary tables uploaded but not indicated in the text.

Reviewers' comments:

Reviewer's Responses to Questions

**Comments to the Author**

1. If the authors have adequately addressed your comments raised in a previous round of review and you feel that this manuscript is now acceptable for publication, you may indicate that here to bypass the “Comments to the Author” section, enter your conflict of interest statement in the “Confidential to Editor” section, and submit your "Accept" recommendation.

Reviewer #1: All comments have been addressed

Reviewer #2: All comments have been addressed

2. Is the manuscript technically sound, and do the data support the conclusions?

Reviewer #1: Yes

Reviewer #2: Yes

3. Has the statistical analysis been performed appropriately and rigorously? 

Reviewer #1: Yes

Reviewer #2: Yes

4. Have the authors made all data underlying the findings in their manuscript fully available?

Reviewer #1: Yes

Reviewer #2: Yes

5. Is the manuscript presented in an intelligible fashion and written in standard English?

Reviewer #1: Yes

Reviewer #2: No

6. Review Comments to the Author

Reviewer #1: all the recommendations have been addressed properly. i suggest this manuscript is acceptable for publication.

Reviewer #2: Comments

I am satisfied with the response and the revision of authors, however, I suggest to make a couple of corrections:

1. As authors’ results did not indicate a role for CH1 and PAI mRNA of blood EVs in early detection of DN, would it be better to modify the title of the manuscript to be more relevant to the results?

2. I would recommend some close proofreading; some punctuation still needed, and the reference number should be placed in the end of the sentence before the full stop not after.

7. PLOS authors have the option to publish the peer review history of their article (what does this mean?). If published, this will include your full peer review and any attached files.

Reviewer #1: **Yes: **Syeda Zahra Abbas

Reviewer #2: No

---

## [Author Response · Author response to Decision Letter 1]

22 Jan 2022

Response to Reviewers

Reviewer #2: Comments

I am satisfied with the response and the revision of authors, however, I suggest to make a couple of corrections:

1. As authors’ results did not indicate a role for CH1 and PAI mRNA of blood EVs in early detection of DN, would it be better to modify the title of the manuscript to be more relevant to the results?

Done. You are right. We modified the title in the revised manuscript.

2. I would recommend some close proofreading; some punctuation still needed, and the reference number should be placed in the end of the sentence before the full stop not after.

Done. Thanks for your accurate comment. We corrected the punctuation and other grammatical errors to the best of our knowledge.

---

## [Editor Report · Decision Letter 2]

14 Feb 2022

PONE-D-21-29236R2The role of CDH2 and MCP-1 mRNAs of blood extracellular vesicles in predicting early-stage diabetic nephropathyPLOS ONE

Dear Dr. Hashemi,

Thank you for submitting your manuscript to PLOS ONE. After careful consideration, we feel that it has merit but does not fully meet PLOS ONE’s publication criteria as it currently stands. Therefore, we invite you to submit a revised version of the manuscript that addresses the points raised during the review process.

We look forward to receiving your revised manuscript.

Kind regards,

Muhammad Tarek Abdel Ghafar, M.D

Academic Editor

PLOS ONE

Journal Requirements:

Additional Editor Comments:

There are several structural, grammatical, punctuation, and capitalization errors that require heavy editing by a native English speaker. Please revise the organization of the tables as there are supplementary tables uploaded but not indicated in the text.
---

## [Author Response · Author response to Decision Letter 2]

27 Feb 2022

Response to Reviewers:

Additional Editor Comments:

There are several structural, grammatical, punctuation, and capitalization errors that require heavy editing by a native English speaker. Please revise the organization of the tables as there are supplementary tables uploaded but not indicated in the text.

Done, Thanks for your comment. A native English speaker has heavily edited the final version of this manuscript in terms of structural, grammatical, punctuation, and capitalization errors. We also organized all the tables in the revised manuscript. In this case, we had 8 tables in the revised manuscript that were cited in ascending numeric order upon first appearance in the manuscript file from Table 1 to Table 8. Besides, we provided the raw data of participants as S1 Appendix.

---

## [Editor Report · Decision Letter 3]

7 Mar 2022

The role of CDH2 and MCP-1 mRNAs of blood extracellular vesicles in predicting early-stage diabetic nephropathy

PONE-D-21-29236R3

Dear Dr. Hashemi,

We’re pleased to inform you that your manuscript has been judged scientifically suitable for publication and will be formally accepted for publication once it meets all outstanding technical requirements.

Kind regards,

Muhammad Tarek Abdel Ghafar, M.D

Academic Editor

PLOS ONE
---

## [Editor Report · Acceptance letter]

22 Mar 2022

PONE-D-21-29236R3 

The role of CDH2 and MCP-1 mRNAs of blood extracellular vesicles in predicting early-stage diabetic nephropathy 

Dear Dr. Hashemi:

I'm pleased to inform you that your manuscript has been deemed suitable for publication in PLOS ONE. Congratulations! Your manuscript is now with our production department. 

Kind regards, 

on behalf of

Prof Muhammad Tarek Abdel Ghafar 

Academic Editor

PLOS ONE